# The Influence of Microstructure on the Mechanical Properties and Fracture Behavior of Medium Mn Steels at Different Strain Rates

**DOI:** 10.3390/ma12244228

**Published:** 2019-12-17

**Authors:** Zheng Wang, Juanping Xu, Yu Yan, Jinxu Li

**Affiliations:** Corrosion and Protection Center, Institute for Advanced Materials and Technology, University of Science and Technology Beijing, Beijing 100083, China; swangzheng@163.com (Z.W.);

**Keywords:** medium Mn steels, strain rate, δ-ferrite, mechanical properties, fracture behavior

## Abstract

The primary task of automotive industry materials is to guarantee passengers’ safety during a car crash. To simulate a car crash, the influence of strain rates on mechanical properties and fracture behavior of medium Mn steels with different Si content (0Si without δ-ferrite and 0.6Si with about 20% δ-ferrite) was conducted using the uniaxial tensile test. The results show that ultimate tensile strength is higher, whereas total elongation is lower in 0Si than in 0.6Si. As the strain rate increases, ultimate tensile strength and total elongation decrease in both 0Si and 0.6Si; nonetheless, total elongation of 0.6Si decreases faster. Meanwhile, the area reduction of 0.6Si increases as the strain rate increases. The microcrack′s number on a rolling direction (RD)-transverse direction (TD) surface is considerably increased; nonetheless, the microcrack′s size is restrained in 0.6Si compared with 0Si. Microcracks start at γ(α′)/α-ferrite interfaces in both 0Si and 0.6Si, whereas little nucleation sites have also been found at (γ(α′)+α-ferrite)/δ-ferrite boundaries in 0.6Si. Meanwhile, δ-ferrite reveals a higher capacity for microcrack arrest. As the strain rate decreases, increased lower crack growth results in fine and even dimples on fractographs with abundant second cracks on fractographs; meanwhile, the small microcrack′s number increases, while the large microcrack′s number decreases on an RD-TD surface.

## 1. Introduction

The automotive industry requires materials that not only satisfy the need to be lightweight to meet the demand for energy conservation and environmental protection, but can also withstand impact during a car crash to guarantee the safety of passengers [1,2]. As a third-generation advanced high-strength steel (3rd-generation AHSS), medium Mn steels with a mixture of α-ferrite and reverted γ [3,4,5], and medium Mn steels exhibit excellent strength and high ductility owing to the transformation-induced plasticity (TRIP) [6,7,8,9]. The TRIP can be influenced by heat treatment [10], chemical composition [11], and cold deformation [12]. In addition to the TRIP effect, dislocation strengthening and refinement strengthening can also influence the mechanical properties in medium Mn steels [13].

Medium Mn steels with Si have been investigated by some researchers [3,14,15,16], but the purpose of Si addition has rarely been mentioned. It is well known that Si and Al addition can increase the reverted γ transition temperature, which can shorten the intercritical annealing time [17]. Si and Al are the ferrite-forming elements. Hence, Si and Al addition can promote the formation of δ-ferrite in medium Mn steels. Si is also an effective solid solution strengthening element [18]. Si addition can be beneficial for the processing and mechanical properties of medium Mn steels [19]. Sun et al. [19] reported that total elongation (TEL) reached a maximum value at the addition of 1 wt.% Si, while mechanical properties were worse without the added Si. In addition, plenty of coarse δ-ferrite could be gained at the addition of 3 wt.% Si, resulting in poor mechanical properties [19]. However, Cai et al. [20] obtained the best TEL when the microstructure contained plenty of δ-ferrite, demonstrating that results from different studies are often controversial. Meanwhile, other researchers have shown that δ-ferrite can sustain more strain during deformation owing to the strain partitioning [21,22]. The effect of strain partitioning of δ-ferrite can contribute to the high mechanical stability of reverted γ during deformation. However, Sun et al. [19] revealed that the micro-strain of δ-ferrite approximately equals that of α+γ. Namely, there is no strain partitioning of δ-ferrite.

The fracture behavior of medium Mn steels has been investigated by some researchers. Choi et al. [23] identified void nucleation sites in both the δ-ferrite and the interface of δ/α+γ as nonmetallic inclusions. Void nucleation sites at the interface of γ(α′)/α resulted from the strain gradients. The propagation of cracks was observed both within the δ-ferrite and along the δ/α+γ interface parallel to the rolling direction (RD), which is consistent with the analysis by Tonizzo et al. [24]. Although Sun et al. [25] reported the same void nucleation sites, cracks propagated preferentially into the brittle δ-ferrite perpendicular to the RD in both hot and cold rolled 3Si samples. The brittle δ-ferrite resulted from the intermetallic compound of B2 phase. In addition, γ(α′)+α layers show a higher capacity for crack arrest in cold-rolled 3Si samples. However, δ-ferrite was considered a soft phase in medium Mn steels [20,21], which contradicts the result of Sun et al. Zhang et al. [26] revealed that the propagation of cracks was slightly deflected when the cracks met the soft δ-ferrite phase. On the basis of these results, the influence of δ-ferrite resulting from the interaction of Si and Al on fracture behavior is not yet clear.

Strain rate dependent TRIP effects, mechanical properties, and fracture behavior are vital factors during car crashes. However, the influence of the strain rate on these characteristics in medium Mn steels has been rarely described. Firstly, for TRIP effects and mechanical properties, Li et al. [1] demonstrated that the strain rate strongly influences both the TRIP effect and dislocation slip using the nanoindentation method. The dislocation density and strain-induced martensite increased as the strain rate increased. Cai et al. [27] demonstrated that strain-induced martensite decreased as the strain rate increased using the tensile test, a finding that contrasts the result of Li et al. It also revealed that the effect of strain rate (ranging from 1 × 10^−4^/s to 1 × 10^−1^/s) on flow stress was negligible. The TEL decreases as the strain rate increases. Apparently, the TRIP effects and mechanical properties are not yet clear. Second, although fracture behavior has been studied by some researchers [23,24,25], the influence of strain rate on fracture behavior in medium Mn steels has not been mentioned. However, the influence of strain rate on fracture behavior has been investigated in TRIP and dual phase (DP) steels [28,29]. Kim et al. [28] revealed that the plastic neck propagates to neighboring locations at a high strain rate in DP 780 and TRIP 780. Namely, the area reduction decreases as the strain rate increases. Furthermore, regardless of the strain rate, the fractographs exhibit small, round dimples. Cao et al. [29] has reported that the area reduction of DP 800 increases as the strain rate increases at room temperature. Apparently, the results of Kim and Cao are contradictory. However, the documented strain rate effects on TRIP and DP steels cannot be perfectly adapted to medium Mn steels because of the complex microstructure and deformation mechanism. Meanwhile, the effect of the strain rate on nucleation and propagation of cracks has not been mentioned for medium Mn steels. Consequently, it is worthwhile to systematically investigate the effect of strain rate on the fracture behavior of medium Mn steels.

In summary, the influence of strain rate and Si on the mechanical properties and fracture behavior of medium Mn steels is not yet clear. Consequently, the present work aims to investigate the influence of strain rate on the mechanical properties and fracture behavior of 0.2C-6Mn-3Al-(0, 0.6)Si medium Mn steels. The method simulates a car crash [30].

## 2. Materials and Methods

The chemical compositions of the steels are listed in Table 1, and the steels are labeled as 0Si and 0.6Si. Briefly, 21 kg ingots of the steels were cast in a vacuum induction furnace. The ranges of Ac1-Ac3 for both 0Si and 0.6Si are 678–966 °C and 677–1005 °C, respectively. Ac1 and Ac3 were calculated by the Thermo-Calc software with a TCFE7 database (provided by the CISRI-TCS Joint Open Laboratory). The ingots were forged into billets with a section dimension of 100 × 30 mm^2^ after heat treatment for 2 h at 1200 °C and air cooling to room temperature (RT). The billets were solution treated for 2 h at 1200 °C again, hot-rolled to 4.5 mm thick plates via six passes at rolling temperatures from approximately 850 °C to 1150 °C, and then air cooled to RT. Intercritical annealing of both 0Si and 0.6Si was performed at 740 °C for 10 min and 30 min, respectively, obtaining the desired mechanical properties. The samples were then air cooled to RT.

MTS Landmark^®^ (MTS, Eden prairie, MN, USA) servohydraulic test system samples were machined parallel to the rolling direction with a gauge section of 25 × 5 × 1 mm^3^. Uniaxial tensile tests were performed at different strain rates ranging from 10^−4^/s to 10^−1^/s at RT. MTS specimens were mechanically polished before the uniaxial tensile tests.

The microstructure of the specimens, which were etched with 4% nitric acid, was characterized by optical microscopy (OM; Olympus Corporation, Tokyo, Japan). Specimens for electron backscattered diffraction (EBSD; channel 5, Oxford Instruments NanoAnalysis, High Wycombe, UK) measurements were polished mechanically first and then electropolished at RT at a voltage of 25 V for 25 s. The electrolyte consists of perchloric acid and acetic acid in a ratio of 1:9. The accelerating voltage, working distance, and step of EBSD were 20 kV, 17 mm, and 60 nm, respectively. Transmission electron microscopy (TEM) analysis was performed using a TECNAI G2 20 (FEI, Hillsboro, OR, USA) at an accelerating voltage of 200 kV. TEM specimens were mechanically thinned to 50–80 μm, and discs with a diameter of 3 mm were then stamped out. The discs were further thinned using a twinjet electropolisher (RL-1, Rui Ling Chuang Xin, Beijing, China) at 30 V and −30 °C with an electrolyte solution of 5% perchloric acid and 95% alcohol. The fractographs and fracture surface of RD-transverse direction (TD) were characterized by a secondary electron microstructure (SEM; Zeiss, sigma300, Oberkochen, Germany).

The X-ray diffraction (XRD; TTRIII, Riguka, Japan) samples underwent continuous scanning at 5°/min in the 2θ range of 40° to 100°. The volume fraction of reverted γ was calculated as follows [31,32]. The dislocation densities of α-ferrite and newly generated martensite (NGM) were calculated by the classical method of Williamson–Hall (WH) [33,34]. The specimens were step-scanned at 0.004°/s for the peaks (110)α, (200)α, (211)α, and (220)α. Nanoindentation tests (Nano Indenter XP, Keysight Technologies, Germany) were carried out with the cube’s corner tip up to a maximum depth of 2000 nm to measure hardness. The method for producing XRD and nanoindentation specimens was consistent with that used for EBSD specimens.

## 3. Results

### 3.1. Microstructure Characteristics

The influence of Si on the microstructure of medium Mn steel is illustrated in Figure 1, and the specimens were imaged using OM (a–d) and EBSD (e and f). Briefly, 0Si consists of submicron reverted γ + α-ferrite with a hardness of 3.44 GPa (Figure 1a,c,e). However, 0.6Si possesses a certain fraction of coarse-grained δ-ferrite (approximately 20%) with a hardness of 2.90 GPa, which is parallel to the RD (Figure 1b,d) The hardness of submicron reverted γ+α-ferrite of 0.6Si is 3.55 GPa (Figure 1f). The morphology of reverted γ in both 0Si and 0.6Si is mainly lathy (Figure 1e,f).

The volume fraction (XRD), grain size (EBSD), and the concentration of Mn (TEM-Energy dispersive x-ray spectrometry (EDX)) of reverted γ in 0Si and 0.6Si are presented in Table 2. No differences in the volume fraction and Mn contents of reverted γ for both 0Si and 0.6Si are noted. However, the intercritical annealing time of 0.6Si extended to 30 min, which resulted in a larger grain size for the reverted γ (280 nm).

### 3.2. Mechanical Properties

The mechanical properties and strain-induced transformation (SIT) behaviors of both 0Si and 0.6Si are shown in Figure 2. Obviously, the strain rate significantly influences the mechanical properties, especially the TEL (Figure 2a,b). Hence, changing the strain rate is an effective method to control mechanical properties. Figure 2c,d summarize the ultimate tensile strength (UTS), yield strength (YS), and TEL at different strain rates in both 0Si and 0.6Si. Apparently, UTS and YS of 0Si are higher than those of 0.6Si, while TEL is lower in 0Si than in 0.6Si. As the strain rate increases, UTS of 0Si decreases after the initial increase and reaches a maximum (1000 MPa) when the strain rate is 10^−3^/s. With the increasing strain rate, UTS of 0.6Si and TEL in both 0Si and 0.6Si decrease continuously, and YS in both 0Si and 0.6Si increase slightly. Figure 2e displays the transformation ratio of reverted γ in both 0Si and 0.6Si, which shows a similar downtrend as the strain rate increases. The similar downtrend represents an increase in the stability of reverted γ and a similar stability of reverted γ at the same strain rate. The work hardening (WH) rate at different strain rates in both 0Si and 0.6Si is shown in Figure 2f,g. Regardless of the strain rate, WH decreases quickly at the stage of elastic strain. As shown in Figure 2f, WH shows a monotonic decrease after yield until fracture at strain rates of 10^−1^/s and 10^−2^/s. However, WH consists of another stage, which is characterized by a fluctuation of WH after a monotonic decrease at strain rates of 10^−3^/s and 10^−4^/s. This phenomenon represents a dramatic SIT, which can postpone necking and promote a large TEL. This fluctuation is increasingly noted as the strain rate decreases. In addition, the intensity of the fluctuation corresponds to the flow stress (magnified in Figure 2a). We determined a critical strain value, ε_c_, at which a shift from continuous flow stress to jerky flow stress is noted during deformation. During the mutation, martensite transformation was activated. The ε_c_ is lowest when the strain rate is 10^−4^/s. No shift is noted at strain rates of 10^−1^/s and 10^−2^/s. Regarding the WH rate of 0.6Si, there is an additional stage (indicated by the black arrows in Figure 2g) controlled by the δ-ferrite that can sustain more microscopic strain than macroscopic strain [21,35]. The residual WH of 0.6Si is similar to that of 0Si, which is mainly related to the influence of strain rate on reverted γ stability.

In general, the strain rate has a great influence on the mechanical properties of steel. The ratio of UTS (R_UTS_) and TEL (R_TEL_) at different strain rates to those at a strain rate of 10^−1^/s and the area reduction in both 0Si and 0.6Si were investigated (Figure 3). Regardless of the Si content, R_UTS_ and R_TEL_ decrease as the strain rate increases (Figure 3a,b). R_UTS_ is almost the same in both 0Si and 0.6Si, whereas R_TEL_ is higher in 0.6Si than in 0Si, showing that TEL decreases faster in 0.6Si than in 0Si as the strain rate increases. As shown in Figure 3c, the area reduction of 0Si is independent of the strain rate. However, the area reduction of 0.6Si increases as the strain rate increases. The results strongly suggest that δ-ferrite promotes heterogeneous deformation in the neck region as the strain rate increases. Meanwhile, the area reduction is lower in 0.6Si than in 0Si, except when the strain rate is 10^−1^/s.

### 3.3. Microstructure Evolution

TEM observation was used to analyze the influence of strain rate on microstructure evolution. For both 0Si and 0.6Si, the morphology of reverted γ and α-ferrite is mainly lathy. Meanwhile, undeformed microstructures note minimal dislocations (Figure 4a,b). Deformed microstructures of 0Si at strain rates of 10^−1^/s and 10^−4^/s are shown in Figure 4c,e, respectively. Regardless of the strain rate, the fractured samples express a high density of dislocations. Deformed microstructures of 0.6Si at strain rates of 10^−1^/s and 10^−4^/s are shown in Figure 4d,f, respectively. Apparently, regardless of the strain rate, the dislocation density is much higher in 0.6Si than in 0Si, and the dislocation density is slightly higher in 0.6Si at a strain rate of 10^−1^/s. These results strongly indicate that the deformed microstructure of reverted γ+α-ferrite is similar, except for the higher dislocation density in 0.6Si.

Compared with 0Si, it is worthwhile to investigate the microstructure evolution of δ-ferrite in 0.6Si at different strain rates. As shown in Figure 5a, undeformed microstructures exhibit few dislocations. Figure 5b shows the selected area electron diffraction (SAED) of the area within the red circle in Figure 5a. Although deformed microstructures represent severe dislocation pileup at different strain rates, there are still differences between Figure 5c–h. As shown in Figure 5c, the dislocation pile-up is more uneven than that in Figure 5f. To investigate the dislocation distribution in detail at the different strain rates, local magnified maps were analyzed (Figure 5d,e,g,h). Apparently, the dislocation distribution on one side of the grain boundary (GB) (area I) is more uneven in Figure 5d than in Figure 5g. For the dislocation interior of δ-ferrite grains, the results reveal that the dislocation distribution is still uneven at a strain rate of 10^−1^/s (area II in Figure 5e). As shown in Figure 5h, the dislocation sites distributed and piled up homogeneously at a strain rate of 10^−4^/s. In summary, a significant local dislocation pile-up is formed, resulting in an uneven dislocation distribution in δ-ferrite as the strain rate increases.

To quantitatively explore the relationship between the dislocation density and the strain rate, XRD analyses of fractured 0Si and 0.6Si were conducted. As shown in Figure 6, the dislocation density increases as the strain rate increases, and the dislocation density is higher in 0.6Si than in 0Si after fracture, which is consistent with the results in Figure 4. The Orowan equation [36] reveals that the velocity of dislocation motions increase as the strain rate increases. According to the theory of classical mechanics, short-range resistance increases as the velocity of dislocation motions increase [36,37], which results in a significant local dislocation pile-up (Figure 5c–e). Consequently, the dislocation density increases as the strain rate increases. The major deformation mechanism of 0Si is SIT. In addition to SIT, 0.6Si consists of plenty of δ-ferrite, and the deformation mechanism of δ-ferrite is dislocation glide owing to the high stacking fault energy (Figure 5). Meanwhile, the TEL is higher in 0.6Si than in 0Si (Figure 2d). Therefore, the dislocation density of 0.6Si is higher than that of 0Si after fracture.

### 3.4. Fracture Behavior

To investigate the behavior of the steel at different strain rates, fractographs of both 0Si and 0.6Si were observed by SEM (Figure 7). Regardless of the strain rate, 0Si undergoes obvious necking (Figure 7a1,b1), while the area reduction is independent of the strain rate (Figure 3c). However, as shown in Figure 7c1,d1, the necking trend for 0.6Si increases as the strain rate increases, which is consistent with the trend of area reduction (Figure 3c). For both 0Si and 0.6Si, the number of second cracks increases as the strain rate decreases. Meanwhile, there are more second cracks in 0.6Si than 0Si when the strain rate is 10^−4^/s. Although the specimens exhibit ductile fractographs characterized by dimples at different strain rates, some differences are noted (Figure 7a2–d2). The size of the dimples, characterized by diameter and depth, is large and uneven when the strain rate is 10^−1^/s (Figure 7a2,c2). Nevertheless, dimple size tends to decrease and is well proportioned at a strain rate of 10^−4^/s (Figure 7b2,d2).

Figure 8 shows the cross-sectional (RD-TD) SEM images for both 0Si and 0.6Si at different strain rates. The specimens were directly etched after fracture. The image in the bottom left corner is a local magnification area. As shown in Figure 8a, 0Si consists of some large microcracks (yellow circles) vertical to the RD and a few small microcracks at the interface of γ(α′)/α-ferrite (red circles) at a strain rate of 10^−1^/s. As shown in Figure 8b, the number and size of large microcracks decrease (yellow circles that are magnified in the top right corner). However, small microcracks are similar to those of Figure 8a (red circles) when the strain rate is 10^−4^/s. The number of microcracks is considerably increased in 0.6Si compared with 0Si (Figure 8c,d). There are many large microcracks (yellow circles) with a few small microcracks (red circles) at a strain rate of 10^−1^/s (Figure 8c). As shown in Figure 8d, numerous small microcracks (red circles) with a few large microcracks (yellow circles) are noted at a strain rate of 10^−4^/s. Clearly, although the strain rate affected the relative number of large and small microcracks, the nucleation and propagation of microcracks are almost the same.

To investigate the influence of Si on the nucleation and propagation of microcracks, Figure 9 reveals the results of the fractured cross sectional (RD-TD) SEM investigation in both 0Si and 0.6Si. The specimens were first mechanically polished and then etched after fracture. As shown in Figure 9a, the yellow arrows represent the nucleation sites of microcracks. Apparently, the interfaces of γ(α′)/α-ferrite are the major nucleation sites of microcracks in 0Si. For 0.6Si, two nucleation sites were investigated: the interfaces of γ(α′)/α-ferrite (Figure 9b) and the boundaries of (γ(α′)+α-ferrite)/δ-ferrite (Figure 9c). Statistical results reveal that most of the microcrack nucleation sites are located in the phase of γ(α′)+α-ferrite. Microcrack propagation continued in the phase of γ(α′)+α-ferrite. However, microcrack arrest occurs, extending to soft δ-ferrite (Figure 9d). Next, microcracks propagated into the soft δ-ferrite, which contributes to the failure of 0.6Si (Figure 9e).

On the basis of these investigations, the cracking behavior and failure process of 0.6Si can be proposed as follows: (a) firstly, microcracks nucleate at the interfaces of γ(α′)/α-ferrite; (b) secondly, microcracks propagate in the phase of γ(α′) and α-ferrite; (c) thirdly, when the microcracks reach the soft δ-ferrite, the microcracks can be arrested; and (d) finally, the microcracks continue to accumulate and grow in the phase of γ(α′) and α-ferrite until propagating to the soft δ-ferrite. The repetition of this process contributes to the fracture of 0.6Si.

## 4. Discussion

### 4.1. The Influence of Si

#### 4.1.1. Microstructure

It is known that C and Mn can stabilize austenite, whereas Al and Si can stabilize ferrite [17]. Apparently, the formation of δ-ferrite is promoted as the Al/Si content increases [19,20]. Sun et al. [19] revealed that 3 wt.% Si added to 0.2C-10Mn-3Al-(0-3)Si steel resulted in a large fraction of δ-ferrite, while there was no δ-ferrite at the addition of 1 wt.% Si. Xu et al. [38] reported almost the same microstructure for 0.18C-6.4Mn-2.8Al-0.5Si steel, whose chemical composition was similar to that of 0.6Si. Consequently, the relative amounts of C, Mn, Al, and Si elements play a crucial role in the microstructure of medium Mn steels. Both Xu et al. [38] and Sun et al. [39] have reported that δ-ferrite could be retained from the hot rolling stage and suffered from recrystallization and growth during intercritical annealing. Compared with the microstructure of 0Si, 0.6Si possesses a certain fraction of coarse-grained δ-ferrite. Consequently, 3 wt.% Al cannot contribute to the formation of the δ-ferrite in 0Si, and the additions of 0.6 wt.% Si and 3 wt.% Al could result in plenty of δ-ferrite.

#### 4.1.2. Mechanical Properties

As shown in Figure 2c,d, in general, UTS is higher in 0Si than in 0.6Si, while TEL is lower in 0Si than in 0.6Si. For UTS, the components of the strengthening of 0Si and 0.6Si mainly consist of dislocation strengthening, phase transformation strengthening, and refinement strengthening during deformation [13]. As shown in Figure 6, the dislocation density is higher in 0.6Si than in 0Si after fracture. Namely, dislocation strengthening is not the primary strengthening mechanism. Consequently, the remaining factors affecting UTS are phase transformation strengthening and refinement strengthening. To simplify analysis of the effect of Si on mechanical properties, both UTS and the instantaneous hardening index (n) were investigated at a strain rate of 10^−4^/s. Although UTS is much higher in 0Si than in 0.6Si (Figure 10a), values of n are almost the same for 0Si and 0.6Si (Figure 10b). These results indicate that phase transformation strengthening is almost the same in both 0Si and 0.6Si during deformation, which is consistent with the results in Figure 2e. Apparently, in addition to submicron-grained reverted γ and α-ferrite, 0.6Si possesses plenty of coarse-grained δ-ferrite (approximately 20%). However, the microstructure of 0Si is submicron-grained reverted γ and α-ferrite. As shown in Figure 10a, YS is much higher in 0Si than in 0.6Si. On the basis of the Hall–Petch equation, it is inferred that refinement strengthening is a major factor resulting in higher YS and UTS in 0Si than in 0.6Si. In addition, the presence of soft δ-ferrite can increase TEL of 0.6Si [20].

#### 4.1.3. Fracture Behavior

As shown in Figure 8, regardless of the strain rate, the number of microcracks is considerably increased in 0.6Si compared with 0Si. The results in Figure 9a reveal that the microcracks of 0Si nucleate at the interfaces of γ(α′)/α-ferrite, which is similar to the findings of Sun et al. [25]. Nucleation of microcracks can be facilitated by the interface decohesion of γ(α′)/α-ferrite, which results from the strain gradients [23] and the low toughness of fresh martensite [25]. Microcrack nucleation at the interfaces of α′ and α-ferrite was widely investigated in dual phase steels [40,41]. For 0.6Si, most of the nucleation of microcracks is consistent with that of 0Si. Nevertheless, there are few nucleation sites at boundaries of (γ(α′)+α-ferrite)/δ-ferrite owing to the large plasticity mismatch [23,25]. However, soft δ-ferrite reveals a higher capacity for crack arrest. Although microcracks can nucleate at the boundaries of (γ(α′)+α-ferrite)/δ-ferrite, the microcracks cannot propagate into the soft δ-ferrite. Microcrack propagation arrest by δ-ferrite results in a smaller crack size in 0.6Si than in 0Si at a strain rate of 10^−1^/s. At a strain rate of 10^−4^/s, the microcrack size is much smaller in both 0Si and 0.6Si, which results from the effect of strain rates. Consequently, δ-ferrite is not a dominant factor in the small microcrack size, leading to similarities in microcrack size for 0Si and 0.6Si at a strain rate of 10^−4^/s.

### 4.2. The Influence of Strain Rate

#### 4.2.1. Ultimate Strength (UTS)

In general, UTS of both 0Si and 0.6Si increases as the strain rate decreases. As shown in Figure 6, the density of dislocation increases as the strain rate increases. The dislocation strengthening increases as the strain rate increases. As shown in Figure 2e, the ratio of phase transformation decreases, which weakens the phase transformation strengthening as the strain rate increases. The above analysis reveals that dislocation strengthening increases and phase transformation strengthening decreases as the strain rate increases. Hence, competition between the two strengthening mechanisms has a decisive effect on UTS during deformation.

To investigate the competition of strengthening mechanisms during deformation, n was assessed. For 0Si, the rate of increase in n is independent of the strain rate at the early stage when the specimens undergo plastic deformation (the red rectangle in Figure 11a). However, as the true strain increases, the three intersections of 1, 2, and 3 revealed that the rate of increase in n increases faster as the strain rate increases (the magnified image of the black rectangle in Figure 11a). Hence, it can be inferred that the rate of dislocation strengthening increases faster as the strain rate increases. However, the rate of increase in n increases faster as the strain rate decreases when the true strain increases continuously (the three intersections of 4, 5, and 6 in Figure 11a). Consequently, the rate of phase transformation strengthening increases faster when the strain rate decreases. The phenomenon for 0.6Si is similar to that of 0Si (Figure 11b). In summary, although dislocation strengthening decreases, phase transformation strengthening is greatly promoted as the strain rate decreases, contributing to the increase of UTS.

#### 4.2.2. Total Elongation (TEL)

As shown in Figure 2d, TEL of both 0Si and 0.6Si decreases as the strain rate increases. M.M. Wang et al. [42] reported that sufficient dislocation glide results in a stacking fault that can cause the reverted γ to transform into martensite. However, the increase in the short-range resistance makes the mean free path of dislocation decrease, which causes severe local pile-up of the dislocation at a high strain rate. This feature can restrain the interaction of dislocation and the reverted γ. Hence, the reverted γ becomes stable, which can weaken the TRIP effect. This situation promotes the onset of necking, and eventually decreases the TEL continuously as the strain rate increases [43]. As the strain rate increases, the restriction of the strain partitioning effect of δ-ferrite [21], resulting from the local uneven dislocation pile-up (Figure 5c–e), can further reduce TEL of 0.6Si.

The phase transformation of reverted γ could be delayed or stopped if the temperature of the specimens increases [28,30,44]. The temperature increase mainly occurs under adiabatic conditions at a high strain rate [29]. The temperature will increase to approximately 60 °C in adiabatic conditions at a strain rate of 10^−1^/s. The result can both promote the annihilation of dislocation and restrain the phase transformation, limiting the TRIP effect that results in the TEL decrease.

#### 4.2.3. Fracture Behavior

The fracture mechanism for engineering materials is the cavity growth mechanism [45]. Stowell [46,47] revealed that cavity growth is mainly controlled by the mechanism of local plastic deformation. The relationship between the rate of cavity growth and plastic deformation is as follows:(1)$$\frac{dv}{dt}=\eta \nu \dot{\varepsilon }$$
where *ν* is the cavity volume, $\frac{dv}{dt}\;$ is the rate of cavity growth, *η* is the parameter of $\frac{dv}{dt}$, and $\dot{\varepsilon }$ is the strain rate. The rate of cavity growth increases as the strain rate increases.

As the strain rate decreases, the microstructures of the specimens are relatively homogeneous during deformation, given the regulation of dislocation. The rate of crack growth is reduced; thus, the cavity has abundant time to nucleate and, as a result, the dimples are fine and even (Figure 7b2,d2). Meanwhile, the dramatic TRIP effect promotes the redistribution of stress concentrations. This situation can alleviate the local stress concentration, resulting in an increase in the density of second cracks (Figure 7b1,d1). Kang et al. [48] also revealed that the formation of delamination on fractographs can promote the occurrence of ductile fracture, contributing to the fine and even dimples at a strain rate of 10^−4^/s.

The number and size of large microcracks on the surface of RD-TD samples decreases as the strain rate decreases (Figure 8). Zhang et al. [26] and Shen et al. [49] demonstrated that delamination could relax the triaxial stress conditions ahead of the advancing crack front. This phenomenon postpones the propagation of microcracks, decreasing the number and size of large microcracks. The number of small microcracks on the surface of RD-TD samples increases as the strain rate decreases. As shown in Figure 2e, the transformation ratio of reverted γ increases, which contributes to a rising TRIP effect, delaying the onset of necking as the strain rate decreases [50]. As discussed in 4.1.3, microcracks nucleate at the interface of γ(α′)/α-ferrite, which results from the strain gradients [23] and the low toughness of fresh martensite [25]. Consequently, more transformation from reverted γ to α′ provides more microcrack nucleation sites as the strain rate decreases.

The area reduction of 0.6Si increases as the strain rate increases. Namely, δ-ferrite promotes heterogeneous deformation at a high strain rate owing to the local uneven dislocation distribution (Figure 5c–e). It makes the strain rate of the neck region become gradually larger than the initially imposed strain rate.

## 5. Conclusions

The influence of small additions of Si on the mechanical properties and fracture behavior of medium Mn steels at different strain rates was investigated in detail. The major conclusions are summarized as follows.
(1)The Si addition contributes to about 20% volume fraction of the coarse-grained δ-ferrite phase in 0.6Si. UTS is higher in 0Si than in 0.6Si owing to strong refinement strengthening, whereas TEL is higher in 0.6Si than in 0Si because of the presence of soft δ-ferrite. As the strain rate increases, UTS and TEL decrease as a result of restriction of the TRIP effect for both 0Si and 0.6Si. Nonetheless, TEL of 0.6Si decreases faster owing to the additional limitation of the strain partitioning effect of δ-ferrite as the strain rate increases. Meanwhile, the area reduction of 0.6Si increases owing to the promotion of heterogeneous deformation resulting from δ-ferrite as the strain rate increases.(2)Regarding the influence of Si on fracture behavior, 0.6Si forms more microcracks, but the microcrack size in 0.6Si is reduced compared with 0Si on the sample surface (RD-TD). The interfaces of γ(α′)/α-ferrite are the major nucleation sites of microcracks in both 0Si and 0.6Si. However, there are few nucleation sites at the boundaries of (γ(α′)+α-ferrite)/δ-ferrite in 0.6Si. The propagation of microcracks is promoted in the phase of γ(α′)+α-ferrite, while the soft δ-ferrite reveals a high capacity for microcrack arrest in 0.6Si.(3)The influences of strain rate on fracture behavior are as follows. As the strain rate decreases, the drastic TRIP effect and the slower speed of crack growth contribute to fine and even dimples and an increase in second crack density on fractographs for both 0Si and 6Si. Meanwhile, as the strain rate decreases, on the RD-TD sample surface, the number and size of large microcracks decreased as a result of relaxation of the triaxial stress conditions ahead of the advancing crack front. The relaxation of stress resulted from delamination. The number of small microcracks, on the RD-TD sample surface, increased because of the promotion of the formation of α′.

## Figures and Tables

**Figure 1 materials-12-04228-f001:**
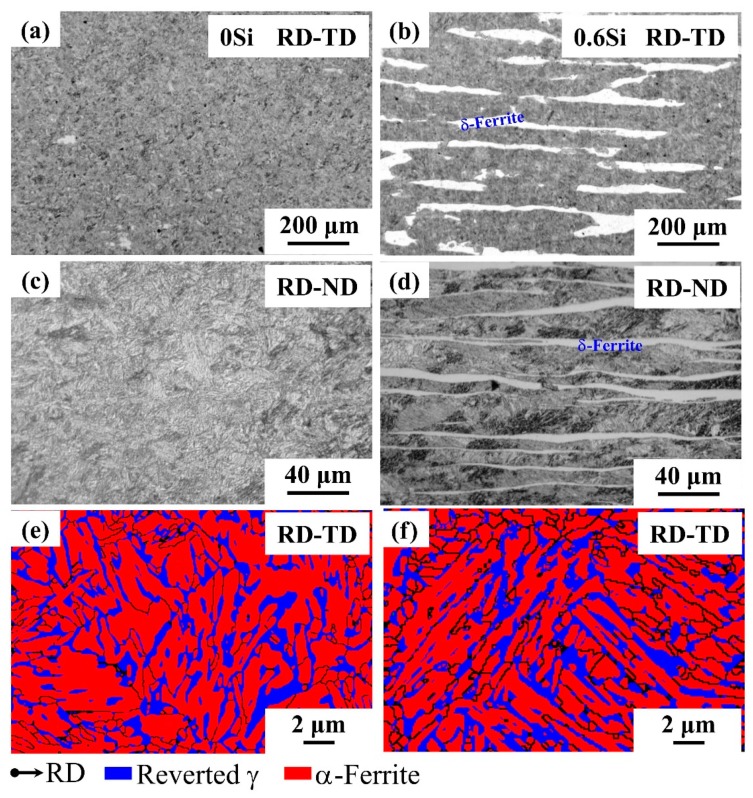
Microstructures of 0Si (**a**,**c**,**e**) and 0.6Si (**b**,**d**,**f**); optical microscopy (OM) microstructures of rolling direction RD-TD (**a**,**b**) and RD-normal direction (ND) (**c**,**d**) and electron backscattered diffraction (EBSD) microstructure of RD-TD (**e**,**f**). The blue and red phases represent the reverted γ and α-ferrite, respectively.

**Figure 2 materials-12-04228-f002:**
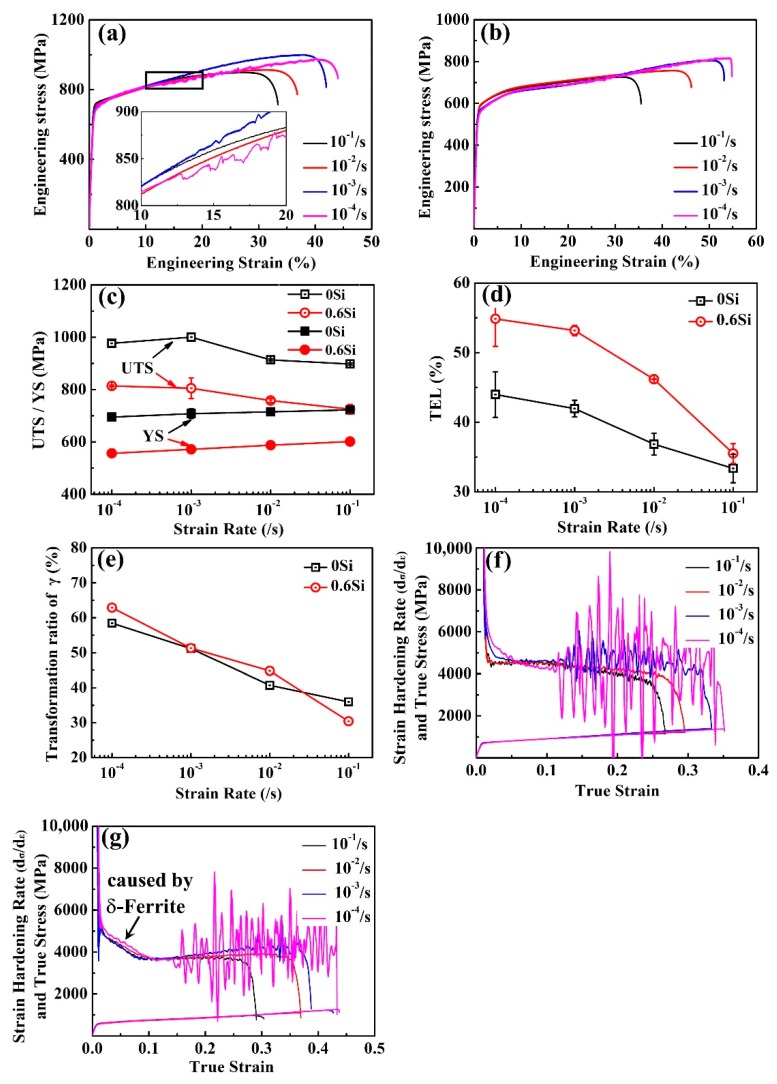
The stress-strain curves of 0Si (**a**) and 0.6Si (**b**); ultimate tensile strength (UTS) and yield strength (YS) (**c**), total elongation (TEL) (**d**), and reverted γ transformation ratio (**e**) at different strain rates in both 0Si and 0.6Si; the true stress-true strain curves and strain hardening rate curves of 0Si (**f**) and 0.6Si (**g**).

**Figure 3 materials-12-04228-f003:**
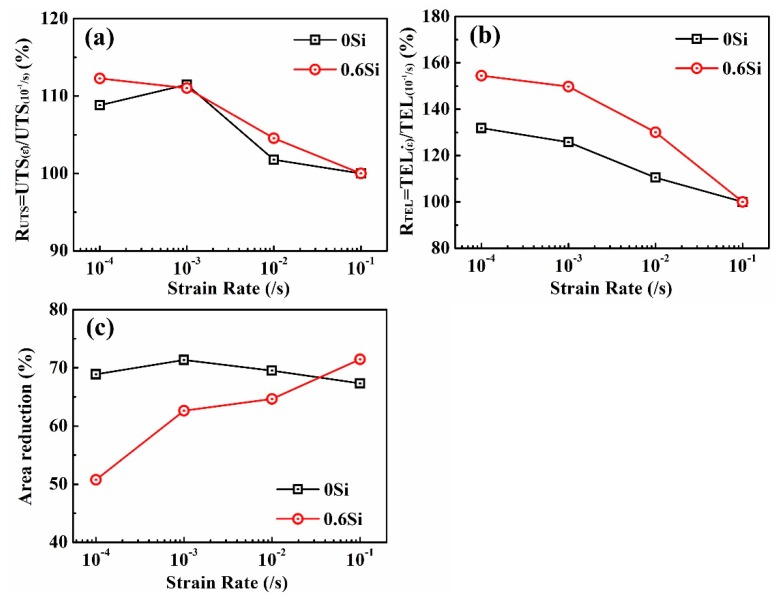
The ratios of UTS (RUTS) (**a**) and TEL (RTEL) (**b**) at different strain rates compared to the ratios at a strain rate of 10^−1^/s vs. strain rates and the area reduction (**c**) vs. strain rates.

**Figure 4 materials-12-04228-f004:**
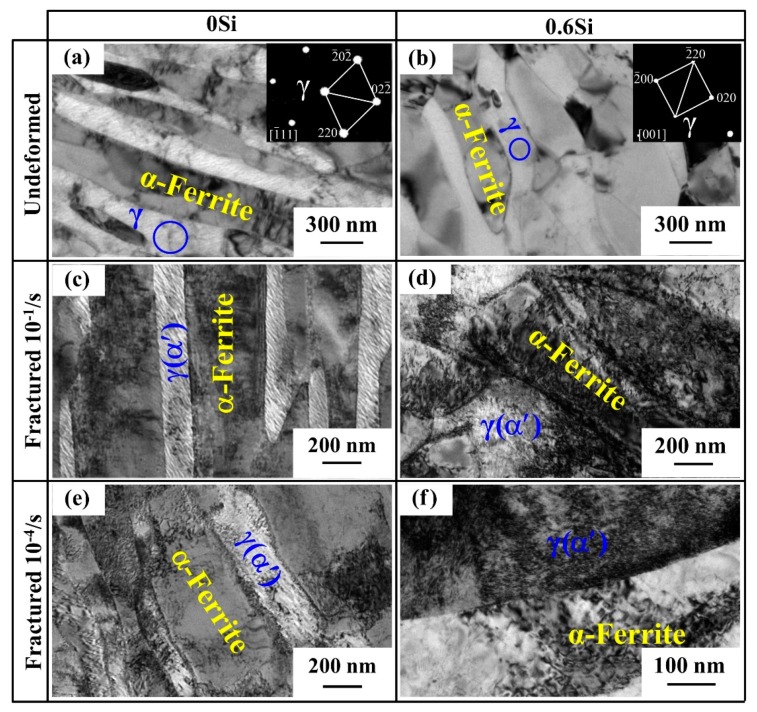
Transmission electron microscopy (TEM) observation of reverted γ+α-ferrite for both 0Si and 0.6Si; 0Si (**a**,**c**,**e**), 0.6Si (**b**,**d**,**f**); undeformed (**a**,**b**) and deformed microstructure of fractured specimens (**c**–**f**) at different strain rates: 10^−1^/s (c and d) and 10^−4^/s (**e**,**f**).

**Figure 5 materials-12-04228-f005:**
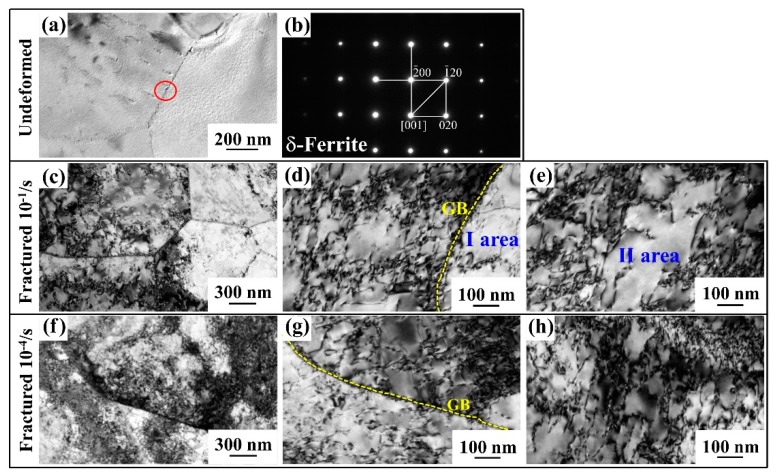
TEM observation of δ-ferrite in 0.6Si: undeformed microstructure (**a**), electron diffraction spot of selected area (black circle in a) (**b**), and deformed microstructure of fractured specimens (**c**–**h**) at different strain rates, 10^−1^/s (**c**–**e**) and 10^−4^/s (**f**–**h**).

**Figure 6 materials-12-04228-f006:**
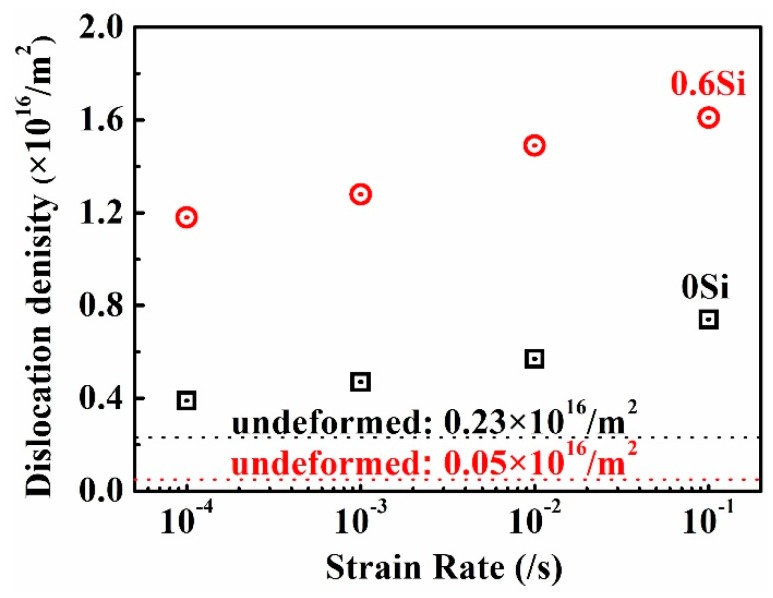
The dislocation density of fractured specimens of both 0Si and 0.6Si at different strain rates; the black and red dotted lines represent the dislocation densities of undeformed 0Si and 0.6Si, respectively.

**Figure 7 materials-12-04228-f007:**
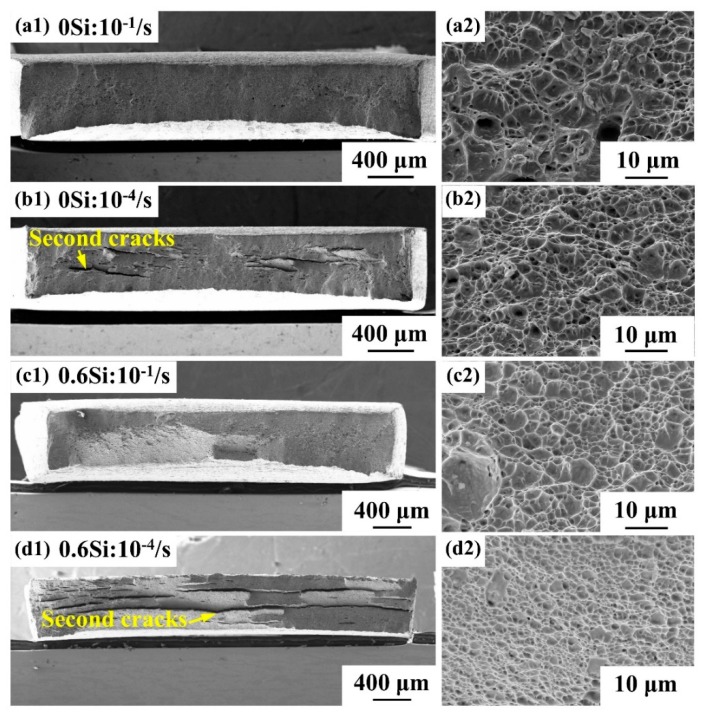
Fractographs at different strain rates, 10^−1^/s (**a**,**c**) and 10^−4^/s (**b**,**d**) for 0Si (**a**,**b**) and 0.6Si (**c**,**d**); macroscopic fractographs (**a1**,**b1**,**c1**,**d1**), and locally magnified fractographs (**a2**,**b2**,**c2**,**d2**).

**Figure 8 materials-12-04228-f008:**
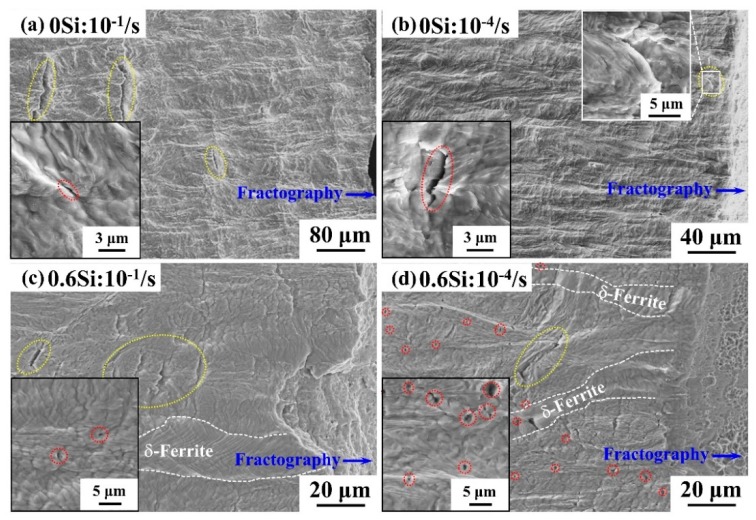
The morphology of the surface of RD-TD at different strain rates, 10^−1^/s (**a**,**c**) and 10^−4^/s (**b**,**d**) for 0Si (**a**,**b**) and 0.6Si (**c**,**d**). The image in the bottom left corner represents a local magnification area. The red circles represent small cracks and the yellow circles represent large cracks.

**Figure 9 materials-12-04228-f009:**
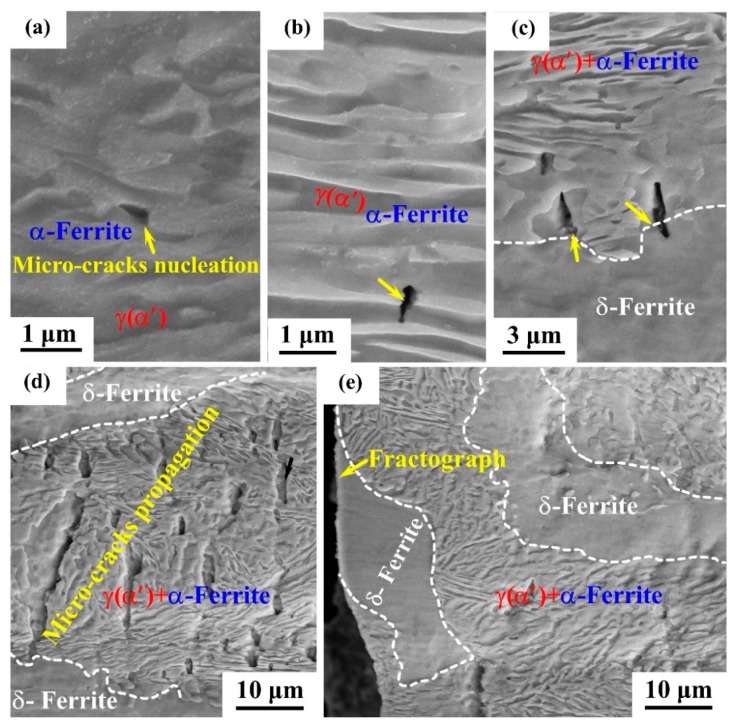
The nucleation and propagation of cracks on the sample’s surface of RD-TD at a strain rate of 10^−4^/s for 0Si (**a**) and 0.6Si (**b**–**e**). The black arrows in (**a**–**c**) represent the nucleation sites of microcracks.

**Figure 10 materials-12-04228-f010:**
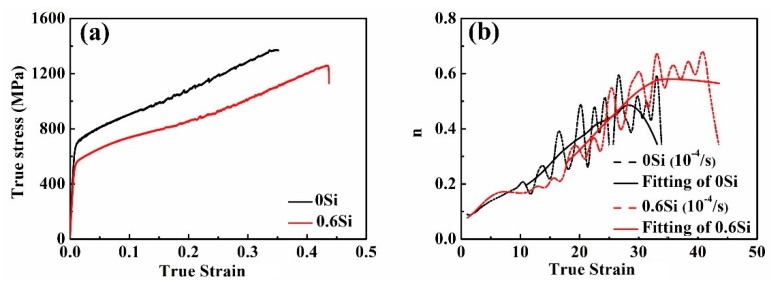
True stress vs. true strain (**a**) and the instantaneous hardening index (n) vs. true strain (**b**) in both 0Si and 0.6Si at a strain rate of 10^−4^/s.

**Figure 11 materials-12-04228-f011:**
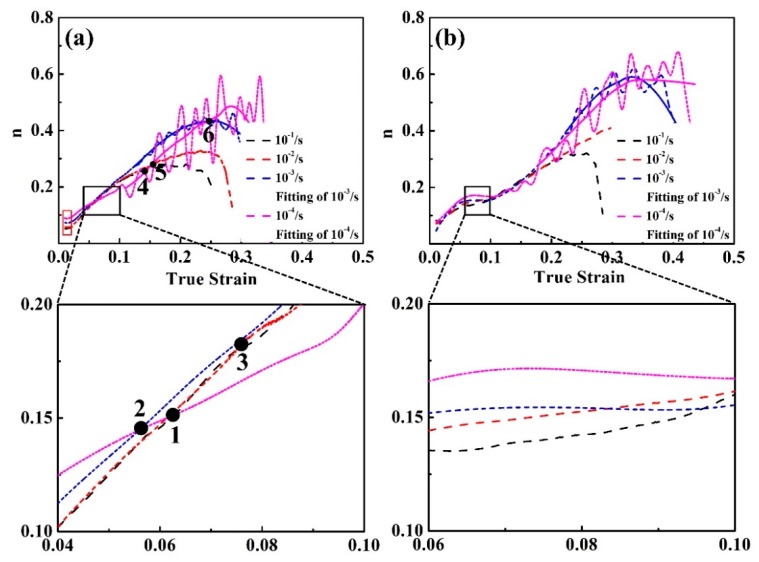
The relationship between the instantaneous hardening index (n) and the true strain at different strain rates of 0Si (**a**) and 0.6Si (**b**).

**Table 1 materials-12-04228-t001:** The chemical compositions of 0Si and 0.6Si (wt.%).

Samples	C	Mn	Al	Si
0Si	0.22	6.12	3.08	–
0.6Si	0.18	6.06	2.87	0.58

**Table 2 materials-12-04228-t002:** Detailed information (volume fraction of reverted γ, Mn concentration in reverted γ, and grain size of reverted γ and α-ferrite) for both 0Si and 0.6Si. XRD, X-ray diffraction; TEM, transmission electron microscopy.

Specimen	Fraction of γ (%)-XRD	Mn Concentration of γ (wt.%)-EDX (TEM)	Grain Size (nm)-EBSD	
			γ	α-Ferrite
0Si	35 ± 2	9.2 ± 0.5	230 ± 180	300 ± 284
0.6Si	36 ± 2	9.0 ± 0.5	280 ± 260	336 ± 316

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
