# Peer review of "The Influence of Microstructure on the Mechanical Properties and Fracture Behavior of Medium Mn Steels at Different Strain Rates"

_materials, 2019, doi:10.3390/ma12244228_

Round 1
Reviewer 1 Report
Introduction:
- (Mandatory) Literature survey and discussion about the topic of delta - ferrite.
Please see review in the attached document.
Best regards

Reviewer 2 Report
The goal of this research is to study the influence of strain rates on the microstructure and mechanical properties of medium Mn steels with different Si contents. The microstructure was studied by SEM, X-Ray diffraction and TEM. The mechanical properties were obtained from tensile tests. In order to research the influence of the strain rate, the experiments have been carried out at different strain rates. The characteristic of the fracture has also been analyzed by SEM.
The work presented in this manuscript is very complete, original and it is very interesting. So, it deserves to be published. I have found the following deficiencies that can be corrected in order to improve the quality of the manuscript:
Line 87. It is said “100x30 mm2”. It should be 100x30 mm2 Line 95. It is said “25x5x1 mm3”. It should be 25x5x1 mm3 Line 96. On writing the strain rates, -4 and -1 they must be 10-4 and 10-1. Lines 114, 115, 116. There must be a mistake, these lines must be eliminated as they do not correspond to the manuscript. Table 2. The number of figures both in the values and in the uncertainties are not correct. In my opinion they should be: 35 (±)2; 36(±2); 9.2(±0.5); 9.0(±0.5); 0.2(±0.2); 0.3(±0.3); 0.3(±0.4); 0.3(±0.4) Figures 2 (a) and (b). I think that they correspond to the engineering stress and engineering strain. It is better if it is written on the x-Title and Y-title. Figures 7c, 7d and 7e. The x-Title is missing. I suppose that it is strain rate Line 140. It is said “while TEL is decreased in 0.6Si”. It is not clear the meaning of this small sentence. TEL is greater for 0.6Si than for 0Si, and in both cases, the values decrease on increasing the strain rate. Lines 152, 153 and 154. The authors speak about a critical strain value εc. The authors should explain a little more about the meaning of this coefficient. Are the work hardening curves (Fig. 2f and 2g) adjusted following some specific model? On the other hand, it would be nice if the authors give the values of these coefficients.
Reviewer 3 Report
Dear authors, your work may be of an interest because of relatively high level of scientific novelty, but the formal presentation must be sighńificantly improved. Below are the reasons of the decision:
1) The title of the paper is “The influence of small additions of Si on the mechanical properties and fracture behavior of medium Mn steels at different strain rates” however, there is only one “small Si addition” investigated. Please take this into account and change the manuscript title accordingly.
2) Please do not use abbreviations in the abstract - this is superfuous as they are used in the main text.
3) The English requires extensive editing - there is a lot grammar errors in the manuscript. Some of the sentences are too long, which makes it difficult to understand their meaning. In many places in the text inappropriate wording of sentences is used. Below are just some examples:
Line 38: Sun et al. [17] reported that total elongation (TEL) reached a maximum value with the addition of 1 wt.% Si, while mechanical properties were worse without the added Si. – better is “Sun et al. [17] reported that total elongation (TEL) reached a maximum value at the addition of 1 wt.% Si, while mechanical properties were worse without the added Si”
Line 42: However, Cai et al [18] obtained the best TEL when the microstructure contained plenty of delta-ferrite, demonstrating that results from different studies are not uniform. Better is: “However, Cai et al [18] obtained the best TEL when the microstructure contained plenty of delta-ferrite, demonstrating that results from different studies are often controversial.”
Line 180 (and further in the text): “Fractured microstructures” – this term is misleading because the microstructure cannot be “fractured”. From the context of the paper it is clear that the microstructure is “deformed” rather than “fractured” . The term “fractured” is commonly used when describing the fractured surfaces after mechanical testing of materials.
Lines 120 – 123: However, 0.6Si possesses a certain fraction of coarse-grained δ-ferrite (approximately 20%) with a hardness of 2.90 GPa, which is parallel to the RD (Fig. 1 b and d), except for the submicron reverted γ+α-ferrite with a hardness of 3.55 GPa 122 (Fig. 1 f). (this is an example of too long sentence – please divide such long sentences into two – three shorter ones, this will make the text much more understandable for the readers)
4) Provide type, manufacturer (name, city, country) of all the devices used in your experiments.
5) Table 2: The ranges of statistical uncertainty of the grain size values are greater than the mean values. Is it right? Can the grain size reach minus values? Please correct.
6) The diagrams in Figs. 2, 11 are too small (including labelling inside the graphs), which makes serious obstacles in reading. Please magnify them.
7) SEM and TEM micrographs - please use color labelling inside the micrographs to increase the visibility of the texts.
After you address all the comments the paper may be recommended for publication. Your reviewer.
Round 2
Reviewer 1 Report
Thank you for the changes.
Best regards
Reviewer 3 Report
Dear authors, your manuscript looks well after revison, hence, I recommended it for publication in the present form. Kind regards. Your reviewer.